# Effect of Ti Content on the Microstructure and Corrosion Resistance of CoCrFeNiTi_x_ High Entropy Alloys Prepared by Laser Cladding

**DOI:** 10.3390/ma13102209

**Published:** 2020-05-12

**Authors:** Xinyang Wang, Qian Liu, Yanbin Huang, Lu Xie, Quan Xu, Tianxiang Zhao

**Affiliations:** 1Equipment Support and Remanufacturing Department, Army Academy of Armored Forces, Beijing 100072, China; lq717460@163.com (Q.L.); hyb1961@126.com (Y.H.); 2School of Mechanical Engineering, University of Science and Technology Beijing, Beijing 100083, China; 3China Satellite Maritime Measurement and Control Department, Jiangyin 214431, China; xuquan1995@126.com (Q.X.); alpha13567531310@163.com (T.Z.)

**Keywords:** high entropy alloy coating, laser cladding, CoCrFeNiTi, first-principles calculation, corrosion resistance

## Abstract

In this paper, CoCrFeNiTi_x_ high entropy alloy (HEA) coatings were prepared on the surface of Q235 steel by laser cladding. The microstructure, microhardness, and corrosion resistance of the coatings were studied. The mechanism of their corrosion resistance was elucidated experimentally and by first-principles calculations. The results show that CoCrFeNiTi_0.1_ adopts a face-centered cubic (FCC) phase, CoCrFeNiTi_0.3_ exhibits an FCC phase and a tetragonal FeCr phase, and CoCrFeNiTi_0.5_ adopts an FCC phase, a tetragonal FeCr phase, and a rhombohedral NiTi phase. The FCC phase, tetragonal FeCr phase, rhombohedral NiTi phase, and hexagonal CoTi phase are all observed in the CoCrFeNiTi_0.7_ HEA. The alloys assume the dendritic structure that is typical of HEAs. Ni and Ti are enriched in the interdendritic regions, whereas Cr and Fe are enriched in the dendrites. With increasing Ti content, the hardness of the cladding layers also increases due to the combined effects of lattice distortion and dispersion strengthening. When exposed to a 3.5 wt.% NaCl solution, pitting corrosion is the main form of corrosion on the CoCrFeNiTi_x_ HEA surfaces. The corrosion current densities of CoCrFeNiTi_x_ HEAs are much lower than those of other HEAs. As the Ti content increases, the corrosion resistance is improved. Through X-ray photoelectron spectroscopy (XPS) and first-principles calculations, the origin of the higher corrosion resistance of the coatings is connected to the presence of a dense passivation film. In summary, the corrosion resistance and mechanical properties of CoCrFeNiTi_0.5_ alloy are much better than the other three groups, which promotes the development of HEA systems with high value for industrial application.

## 1. Introduction

High entropy alloys (HEAs) are composed of five or more elements at equal or nearly equal atomic ratios [1,2,3,4] HEAs tend to form a stable solid solution phase and have excellent mechanical properties and corrosion resistance, unlike the traditional alloys that are typically based on one or two principal elements. Therefore, HEAs have been heavily researched and attracted much attention from the academic community [5,6,7].

The CoCrFeNi HEA system has become important, because of its excellent corrosion resistance, ductility, and thermal stability [8,9]. Surface engineering technologies, such as laser cladding and plasma spraying, can reduce the cost and increase the value of the HEAs for industrial applications [10]. However, the CoCrFeNi HEA system cannot meet the needs of high-strength parts due to its poor mechanical strength. It has been demonstrated that a hard secondary phase can be formed in the CoCrFeNi matrix by adding a small amount of Nb, Ti, or Mo, resulting in precipitation strengthening and improvement of the mechanical properties [11]. For example, Wang et al. showed that a new Lave phase is formed in the CoCrFeNi matrix when a small amount of Nb is added. The formation of the new phase increases the microhardness of the alloy [12]. Praveen et al. studied the effect of Mo addition on microstructure and properties of CoCrFeNi alloy, and found that the crystalline structure changes from a single face-centered cubic (FCC) phase to FCC and σ phase and the hardness increases from 510 Hv to 620 Hv [13]. Previous studies have shown that Ti has a negative mixing enthalpy with Co and Ni. Therefore, it is easy to alloy Ti into CoCrFeNi matrix [14]. Moreover, the addition of Ti is conductive in solid solution strengthening and, hence, improves the mechanical properties of HEAs [15,16]. Löbel et al. added Ti into AlCoCrFeNi alloy and found the hardness and wear resistance of the alloy are increased significantly, which gives it potential for surface protection applications [17]. Besides, the addition of Ti also enhances the corrosion resistance of AlCoCrFeNiCu alloy in the HNO_3_ solution. The corrosion current density of the AlCoCrFeNiCuTi is reduced by 1–2 orders of magnitude, as compared with the substrate [18]. However, the excessive addition of Nb, Ti, or Mo can adversely affect the corrosion resistance of the alloys [19,20] Therefore, there is a need for a systematic study on the effects of various amounts of Ti addition to the CoCrFeNi matrix on the corrosion resistance and mechanical properties of the CoCrFeNiTi_x_ to develop HEA systems with improved properties.

First-principles calculations are based on density functional theory (DFT), and it is widely used in material research [21,22]. Recently, the calculations also have been applied to predict the properties and analyze the mechanisms of HEAs, especially in basic mechanisms during the process of corrosion, which can reduce uncertainties and improve the efficiency in the development of HEA system [23].

In this paper, the CoCrFeNiTi_x_ HEA coatings are prepared on Q235 steel substrate by laser cladding. The effect of Ti content on the microstructure and corrosion resistance in a 3.5 wt.% NaCl solution was studied. Furthermore, the corrosion resistance mechanism was analyzed experimentally and by first-principles calculations.

## 2. Experimental Procedures

Q235 steel was used as the substrate for HEA laser cladding coatings, with a size of 40 mm × 30 mm × 10 mm. Q235 steel is an alloy with a base metal of Fe that contains C (≤0.22%), Si (≤0.35%), Mn (≤1.4%), P (≤0.045%), and S (≤0.05%). Ultrasonication in absolute alcohol removed the impurities on the substrate surface. The CoCrFeNi HEA powder was prepared via a vacuum melting gas atomization method while using metallic powders of Co, Cr, Fe, and Ni (all <99.9 wt.% pure). For the vacuum melting gas atomization process, the protective gas was argon, the atomization temperature was about 1600 °C, atomization pressure was 2.8 MPa, and the vacuum degree was 1 × 10^−2^ Pa. Different amounts of metal powder of Ti (<99.9 wt.% pure) was added to CoCrFeNi HEA powder in a ball mill (ND-L, Xuanhao, Nanjing, China) for 3 h and the ball-to-powder weight ratio is 10:1. Besides, the stearic acid was used as the control agent and the diameter of the ball is 6 mm.

The HEA coatings were prepared by high power semiconductor laser (LDF3000-60, Laserline, Mülheim an der Ruhr, Germany). For the laser cladding process, the laser power was 1.3 kW, the scan speed was 4.5 mm/s, the overlap rate was 30%, the spot diameter was 2 mm, the protective gas was argon, the carrier gas speed was 9 L/min., and the turntable speed was 2 rpm. The samples were cut into small pieces of 10 mm × 10 mm × 10 mm and prepared according to the experimental specifications. The CoCrFeNiTi_x_ HEAs were prepared with different Ti contents, where x = 0.1, 0.3, 0.5, or 0.7 and are abbreviated as T1, T2, T3, or T4 respectively.

The crystal structure of the CoCrFeNiTi_x_ HEAs was determined by X-ray diffraction (XRD, Bruker D8 XRD, Bruker AXS, Karlsruhe, Germany) while using Co-K_α_ radiation with a diffraction angle (2θ) of 0–90°, at a scan rate of 10°/min. Phase determination was conducted while using the Joint Committee on Powder Diffraction Standards (JCPDS) cards. The microstructure and chemical composition of the CoCrFeNiTi_x_ HEA coatings were analyzed by scanning electron microscopy (SEM, NovaNanoSEM450/650, FEI, Hillsboro, OR, USA) and energydispersive X-ray spectroscopy (EDS, FeatureMax X-ray spectrometer, Oxford Instruments, Abingdon, UK), respectively.

The microhardness of CoCrFeNiTi_x_ HEA coatings was measured with a microhardness tester (MICROMET-6030, Buehler, Chicago, IL, USA) under a load of 500 g for 10 s, at different depths. The measurements were performed at 10 different locations on a single column with a 4 mm interval between two test columns. The average value of the 10 measurements was calculated, excluding the maximum and minimum values.

The electrochemical characterization of CoCrFeNiTi_x_ HEAs was performed using an electrochemical workstation (Coste CS350, Corrtest Instruments, Wuhan, China). The experiments were conducted in a 3.5 wt.% NaCl solution at room temperature using a three-electrode configuration. The CoCrFeNiTi_x_-coated steel functioned as the working electrode, a saturated calomel electrode was used as a reference electrode, and a platinum plate was used as the counter electrode. The voltage was swept from −1.5 V–1.5 V at a scan speed of 1 mV/s.

X-ray photoelectron spectroscopy studied the chemical composition of the passivation film on the surface of CoCrFeNiTi_x_ HEA coatings, after the polarization experiment (XPS, ESCALAB 250X_i_, ThermoFisher Scientific, Waltham, MA, USA). The spectra were analyzed while using the Thermo Avantage 5.9 software.

## 3. Results

### 3.1. Microstructure of CoCrFeNiTi_x_ HEA powder

SEM and EDS observed the microscopic morphology and element distribution of CoCrFeNiTi_x_ HEA powder, respectively. T3 group is taken as an example. The powder has good sphericity, the diameter of the particles is from 100 µm to 150 µm, the content of each component element is close to the design value, and the element distribution is uniform, which meets the requirements of cladding, as can be seen in Table 1, Figure 1 and Figure 2.

### 3.2. Microstructure of the CoCrFeNiTi_x_ HEAs

The CoCrFeNiTi_x_ HEA coatings were successfully prepared while using the above powder. The thickness of each coating is approximately 1300 µm. Figure 3 shows microscopic morphology and element distribution in the cross-section of CoCrFeNiTi_0.5_ coating. It can be seen that there is a clear interface between the coating and the substrate. The combination is tight and few defects are observed. The coating shows great metallurgical bonding under the influence of melt convection.

Figure 4 presents the XRD spectra of the CoCrFeNiTi_x_ HEAs. Three distinct characteristic XRD peaks at 2θ = 45°, 50°, and 80° are present for all of the alloys, indicating that the main phase of the alloys is the FCC solid solution phase. The formation of the FCC phase is attributed to the high mixing entropy. The large number and close atomic percentages of the constituent elements in the alloy increase the mixing entropy and reduce the Gibbs free energy of the system, which promotes the formation of a stable solid solution phase. There is no second phase observed in the T1 alloy. The small peak near 2θ = 44° in the spectra for the T2, T3, and T4 alloys is identified as Fe and Cr-rich tetragonal phase (σ phase). Another diffraction peak between the FCC (111) peak (2θ ≈ 45°) and σ phase peak (2θ = 44°) is identified as a rhombohedral Ni_2_Ti phase (R phase) in the T3 and T4 alloys. In addition, small peaks near 2θ = 34° and 2θ = 83° appear in the spectra of the T4 alloy, which is identified as the hexagonal Co_2_Ti Lave phase (L phase).

As the Ti content increases, the intensity of the diffraction peaks of the alloy decreases. This observation is due to the increasing lattice distortion from the mutual diffusion of the constituent elements. The lattice parameter of the CoCrFeNiTi_x_ HEAs increases from 3.491 Å to 3.613 Å due to the lattice distortion that is caused by the addition of Ti, which was calculated based on the XRD data.

The atomic radius of Ti (1.45 Å) is significantly larger than the atom radii of the other constituent elements; *i.e.*, Co (1.25 Å), Cr (1.28 Å), Fe (1.26 Å), and Ni (1.24Å). A large atomic radius difference can exacerbate lattice distortion. The atomic radius difference *δ* is generally used to characterize the extent of the lattice distortion, and it is defined by the following equation [24]:$$\delta =\sqrt{\sum_{i=1}^{N}c_{i}{\left(1-r_{i}/\bar{r}\right)}^{2}}$$
where *r_i_* is the atomic radius of the *i*th constituent element and $\bar{r}$ is the average atomic radius, and *c_i_* is the atomic percentage of the *i*th element. The atomic radius differences of CoCrFeNiTi_x_ HEAs, where x = 0.1, 0.3, 0.5, and 0.7, are *δ* = 0.025, 0.040, 0.050, and 0.057, respectively. Thus, as the Ti content increases, *δ*, the lattice distortion, and the lattice constant also increase.

Figure 5 shows the SEM micrographs of the surface of CoCrFeNiTi_x_ HEAs. All of the alloys have a dendritic structure. The gray domains are the interdendritic regions, and the black regions are the dendritic domains, which are designated by A and B, respectively. EDS measured the chemical composition of the different regions of the alloys, as shown in Table 2. The interdendritic regions are enriched with Ti and Ni, whereas the dendritic regions are enriched with Cr and Fe. The Ti and Ni-rich interdendritic regions are likely composed of intermetallic precipitates, i.e., the hexagonal Co_2_Ti phase and the rhombohedral Ni_2_Ti phase, according to the XRD analysis in Figure 1. The Cr and Fe-rich dendritic regions are primarily composed of the FCC solid solution phase. For the T1 and T2 alloy, where Co_2_Ti and Ni_2_Ti phase were not observed by XRD, the elemental segregation is likely to be due to spinodal decomposition during cooling [20,25]. The increase in Ti content causes the further growth of secondary dendritic arms and the decrease of secondary dendritic arm spacing (SDAS). Furthermore, the element segregation is aslo reduced with increasing Ti content.

### 3.3. Microhardness of the CoCrFeNiTi_x_ HEAs

The microhardness of CoCrFeNiTi_x_ HEA coatings was investigated from its surface to the substrate in order to study the mechanical behavior of the coatings. Figure 6 shows the cross-sectional hardness of the CoCrFeNiTi_x_ HEA coatings. The basic trend of the microhardness curves is essentially the same for the alloys. The hardness reaches a peak between 500–1000 µm from the surface and then drops sharply in the bonding zone. As the Ti content increases, the peak hardness of CoCrFeNiTi_x_ HEA coatings increases from 310 Hv to 830 Hv, which is significantly higher than the hardness of the substrate (122 Hv). The high hardness values are attributed to three different mechanisms. First, when the Ti content, with the larger atomic radius, is increased in the FCC solid solution, the lattice distortion increases, which results in an increase in the hardness. Second, a reduction in the SDAS can improve the hardness and the compressive strength of the alloys, as is the case for most alloys [26,27]. Third, the increase of the Ti content can promote the precipitation of hard phases, which contributes to the overall hardness of the alloys.

### 3.4. Corrosion Performance of CoCrFeNiTi_x_ HEAs

Figure 7 shows the polarization curves of CoCrFeNiTi_x_ HEA coatings in a 3.5 wt.% NaCl solution. The polarization curves show four distinct regimes, i.e., a passivation zone, a transition zone, an activation zone, and an over passivation zone. The electrochemical parameters of CoCrFeNiTi_x_ obtained by a linear fit are shown in Table 3 with other common HEAs for comparison. When compared with other HEAs, CoCrFeNiTi_x_ HEAs have an extended polarization region (∆E) and a lower corrosion current density (i_corr_). These results indicate that CoCrFeNiTi_x_ HEAs have excellent corrosion resistance in a 3.5 wt.% NaCl solution. In addition, as the Ti content increases, the polarization potential (E_corr_) shifts to more positive potentials, the polarization current density decreases, and the length of polarization region increases.

Figure 8 shows the SEM micrographs of the surface of the CoCrFeNiTi_x_ HEAs after the polarization experiment. Many small dish-shaped pits, with a diameter of 1–5 µm, appear on the coating surface. Figure 8b shows that some secondary corrosion pits are formed on the inner walls of the larger corrosion pits. The hierarchical pitting morphology indicates that the main form of corrosion of the CoCrFeNiTi_x_ HEAs, in a 3.5 wt.% NaCl solution, is pitting corrosion.

## 4. Discussion

The polarization experiments show that CoCrFeNiTi_x_ HEAs have excellent comprehensive corrosion resistance in a 3.5 wt.% NaCl solution. The presence of a passivation film is hypothesized to be the cause of the high resistance to corrosion. XPS investigated the chemical composition of the passivation film on the surface of the CoCrFeNiTi_x_ HEAs after the polarization experiment was investigated by in order to confirm this hypothesis. Figure 9 shows the XPS spectra of the passivation film on the CoCrFeNiTi_x_ HEAs. The passivation film is mainly composed of seven elements, i.e., Co, Cr, Fe, Ni, Ti, O, and C. The Ni 2p peaks correspond to Ni and NiO. The Co 2p peaks correspond to Co and CoO. The Fe 2p peaks correspond to Fe and Fe_2_O_3_. The Cr 2p peaks correspond to Cr_2_O_3_. The Ti 2p peaks correspond to TiO_2_. The relative atomic percentages of Cr and Ti, as compared to the other metallic elements, are much higher on the surface of the passivation film than in the bulk material (from EDS analysis). Therefore, the main components of CoCrFeNiTi_x_ passivation film are determined to be TiO_2_ and Cr_2_O_3_.

The mechanisms of the pitting corrosion have not been completely elucidated due to the limitations of the available detection methods. Early research reported that pitting corrosion is caused by the competitive adsorption of atomic oxygen and chlorine. After Cl^−^ ions penetrate the surface of the metal, they become enriched at the metal/passivation film interface. As a result, the crystal lattice of the metal is expanded, which increases the tensile stress of the passivation film and eventually destroys it [28]. Therefore, by designing materials with the ability to resist Cl^−^ ion penetration into the surface and retard its diffusion in the passivation film, the corrosion resistance of alloys can be significantly improved.

The energy barrier of Cl^−^ ion penetration and diffusion was calculated by first-principles methods to study the effect of passivation films on Cl^−^ ion erosion. The energy barrier for the entire erosion process can be divided into two parts. The first (*E*1) is the adsorption energy of Cl^−^ ions into oxygen sites at the surface. The second (*E*2) is the diffusion energy of a Cl^−^ ion from one oxygen site to another. *E*1 and *E*2 can both be written as follows [30]:$$E=E_{tot}(A_{fin})-E_{tot}(A_{ini})$$
where *E* is *E*1 or *E*2, *E_tot_* is the total energy of the state, and *A_fin_* and *A_ini_* are the initial and final states of adsorption or diffusion, respectively. The VASP software was used to calculate the total energy, which employs the plane wave ultra-soft pseudopotential method. The exchange and correlation terms were described with the generalized gradient approximation (GGA) of the Perdew-Burke-Ernzerh (PBE) exchange-correlation function. The cutoff energy was set at 300eV, the energy change was set at 10^−5^ eV, and the K-point grid was set to 3 × 3 × 1. Subsequently, Cr_2_O_3_/TiO_2_ passivation film interface of the CoCrFeNiTi_x_ HEA and the Cr_2_O_3_/Nb_2_O_5_ passivation film interface of the CoCrFeNiNb_x_ HEA [12] were formed and optimized while using the Broyden–Fletcher–Goldfarb–Shanno (BFGS) algorithm.

Figure 10 presents the calculated energy barriers for Cl^−^ ion adsorption and diffusion at Cr_2_O_3_ (a position), interface junction (b position), and TiO_2_/Nb_2_O_5_ (c position). In all of the domains, the Cl^−^ ion diffusion energy is lower than the Cl^−^ ion adsorption energy. Furthermore, the energy barriers at the Cr_2_O_3_/TiO_2_ interfaces are higher than the ones at the Cr_2_O_3_/Nb_2_O_5_ interfaces. Additionally, the energy barriers at the interface junctions are the lowest, which is possibly due to lattice expansion at the junction. These results suggest that the passivation film on CoCrFeNiTi_x_ HEAs strongly resist pitting, which helps to explain their excellent corrosion resistance. Reports of the corrosion resistance of CoCrFeTi_x_CuMnW, AlCoCrFeNi, and Al_0.5_CoCrFeNiCu alloys were also related to their dense passivation films [31,32,33].

## 5. Conclusions

(1)The CoCrFeNiTi_x_ HEA coatings were prepared on the surface of Q235 steel by laser cladding and the coatings were well bonded to the substrate. CoCrFeNiTi_0.1_ adopts an FCC phase, CoCrFeNiTi_0.3_ exhibits an FCC phase and a tetragonal FeCr phase, and CoCrFeNiTi_0.5_ adopts an FCC phase, a tetragonal FeCr phase, and a rhombohedral NiTi phase. The FCC phase, tetragonal FeCr phase, rhombohedral NiTi phase, and a hexagonal CoTi phase are all observed in the CoCrFeNiTi_0.7_ HEA.(2)The alloys assume the dendritic structure that is typical of HEAs. Ni and Ti are enriched in the interdendritic regions, whereas Cr and Fe are enriched in the dendrites.(3)With increasing Ti content, the hardness of the cladding layers also increases due to the combined effects of lattice distortion and dispersion strengthening.(4)In a 3.5 wt.% NaCl solution, pitting corrosion was the main form of corrosion on the CoCrFeNiTi_x_ HEA surfaces. CoCrFeNiTi_x_ HEAs have a higher corrosion potential and lower corrosion current density when compared with other HEAs. As the Ti content increases, the corrosion resistance is improved. Furthermore, the high energy barriers at the Cr_2_O_3_/TiO_2_ interfaces improve the corrosion resistance of the alloys.

## Figures and Tables

**Figure 1 materials-13-02209-f001:**
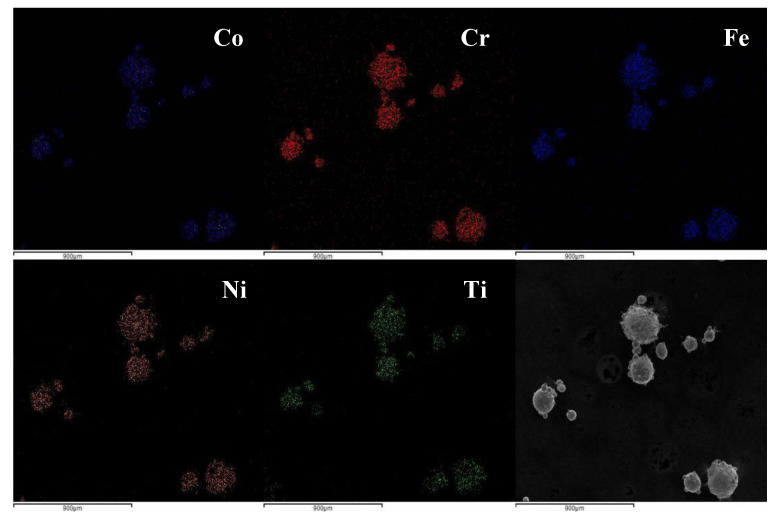
Element distribution of CoCrFeNiTi_0.5_ high entropy alloys (HEA) powder.

**Figure 2 materials-13-02209-f002:**
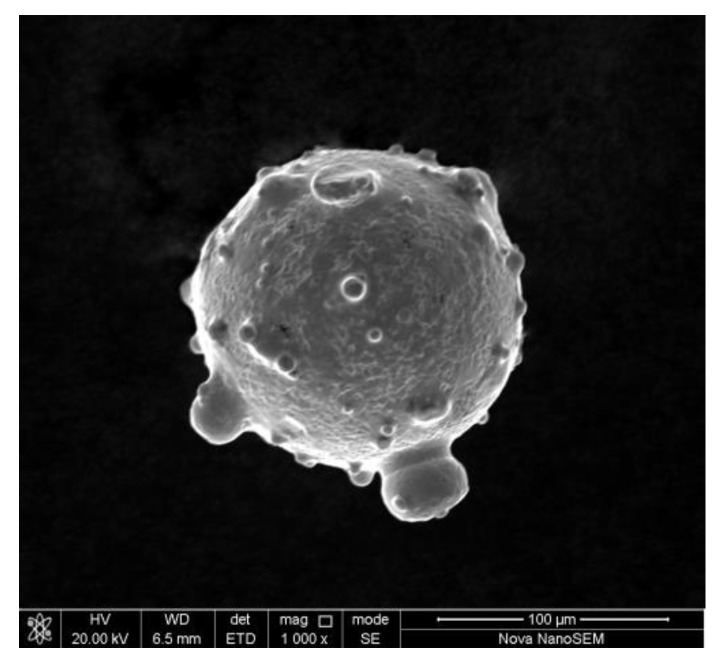
Microstructure of CoCrFeNiTi_0.5_ HEA powder.

**Figure 3 materials-13-02209-f003:**
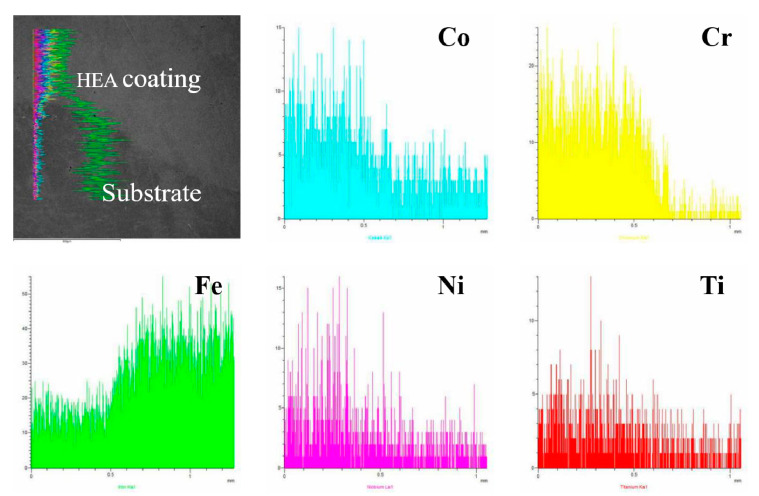
Microscopic morphology and element distribution in the cross-section of CoCrFeNiTi_0.__5_ coating by laser cladding.

**Figure 4 materials-13-02209-f004:**
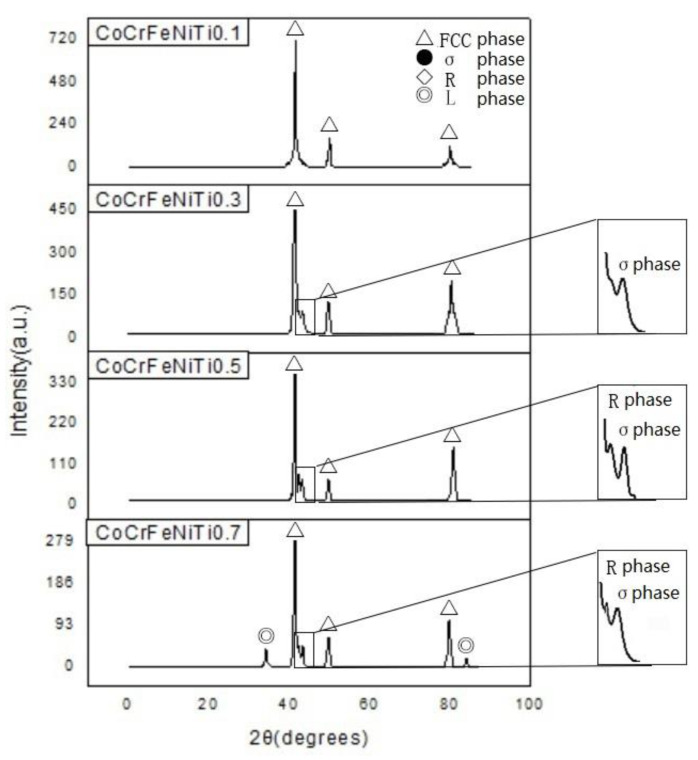
X-ray diffraction (XRD) spectra of CoCrFeNiTi_x_ HEA coatings.

**Figure 5 materials-13-02209-f005:**
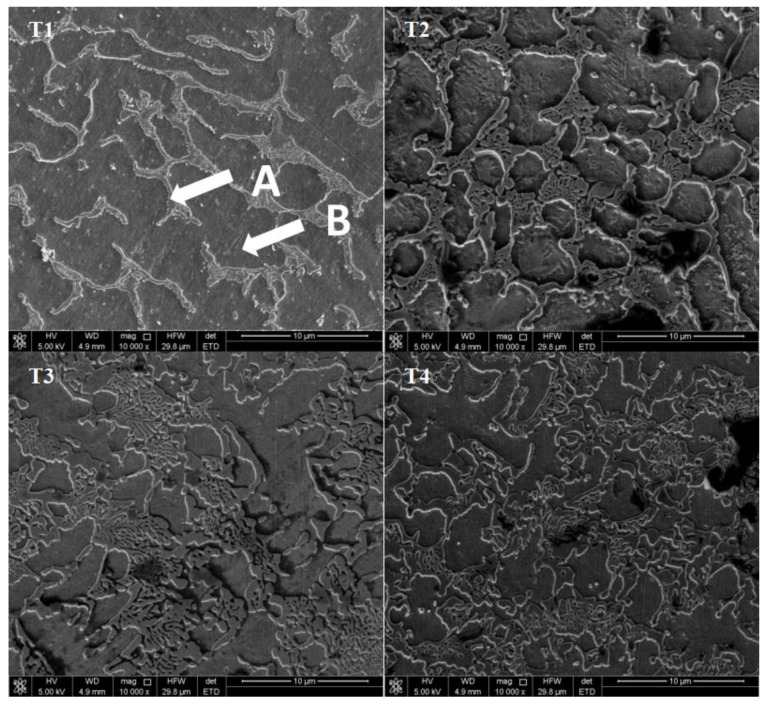
Scanning electron microscopy (SEM) micrographs showing the microstructure of CoCrFeNiTi_x_ HEA laser cladding coatings, where x = 0.1, 0.3, 0.5, or 0.7, and are abbreviated as T1, T2, T3 or T4, respectively.

**Figure 6 materials-13-02209-f006:**
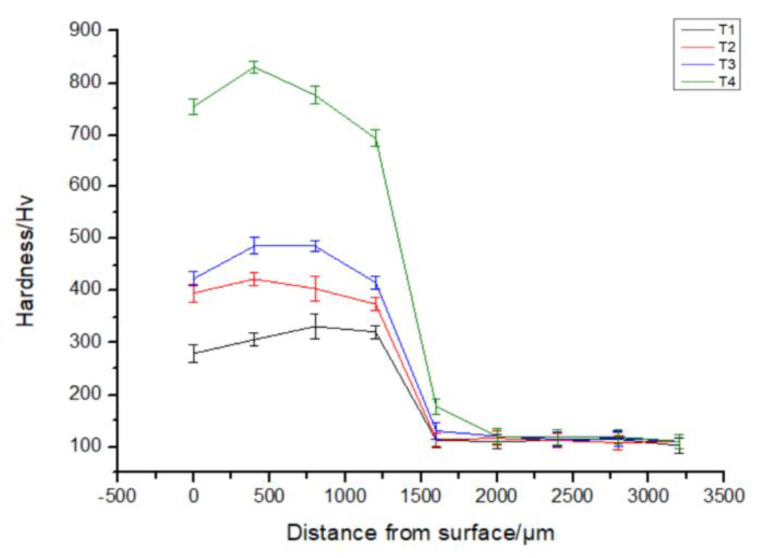
Cross-sectional hardness of CoCrFeNiTi_x_ HEA coatings, where x = 0.1, 0.3, 0.5, or 0.7, and are abbreviated as T1, T2, T3, or T4, respectively.

**Figure 7 materials-13-02209-f007:**
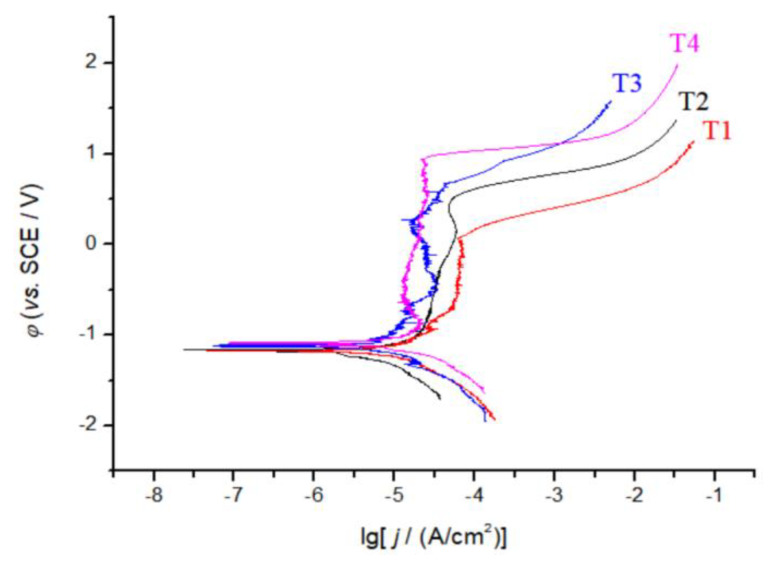
Polarization curves of CoCrFeNiTi_x_ high entropy alloy coatings, where x = 0.1, 0.3, 0.5, or 0.7, and are abbreviated as T1, T2, T3, or T4, respectively, in 3.5 wt.% NaCl solution.

**Figure 8 materials-13-02209-f008:**
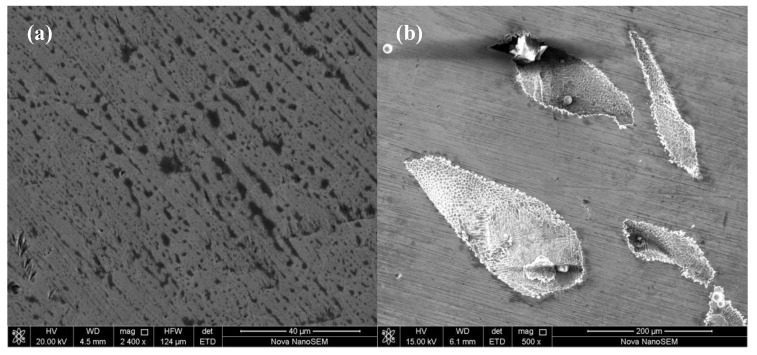
Surface morphology of CoCrFeNiTi_x_ HEA coating after the polarization test: (**a**) low magnification, (**b**) high magnification.

**Figure 9 materials-13-02209-f009:**
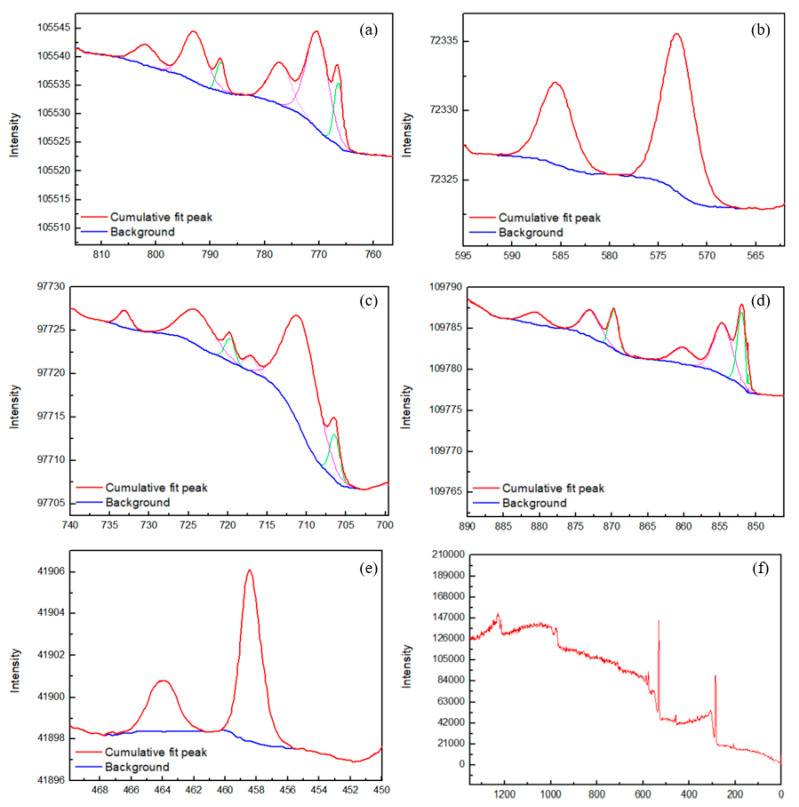
XPS spectra of the passivation film on CoCrFeNiTi_x_ HEA coating. Peaks are attributed to (**a**) Co, (**b**) Cr, (**c**) Fe, (**d**) Ni, and (**e**) Ti species. (**f**) The entire XPS spectral width.

**Figure 10 materials-13-02209-f010:**
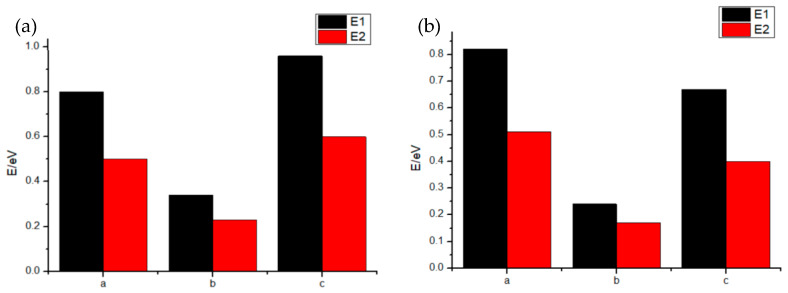
Adsorption energy and diffusion energy of the Cl^−^ at the passivation film of HEAs: (**a**) the results for the Cr_2_O_3_/TiO_2_ interface, (**b**) the results for the Cr_2_O_3_/Nb_2_O_5_ interface.

**Table 1 materials-13-02209-t001:** Element Atomic Ratio of CoCrFeNiTi_0.5_ HEA powder.

Element	Co	Cr	Fe	Ni	Ti
Design value	22.22	22.22	22.22	22.22	11.11
Powder	22.57 (+0.35)	21.25 (−0.97)	22.57 (+0.35)	22.12 (−0.10)	11.48 (+0.37)

**Table 2 materials-13-02209-t002:** Element distribution in the different regions of CoCrFeNiTi_x_ Laser Cladding Coatings, where x = 0.1, 0.3, 0.5, or 0.7, and are abbreviated as T1, T2, T3 or T4, respectively (at.%).

Alloy	Region	Co	Cr	Fe	Ni	Ti
T1	Design value	24.44	24.44	24.44	24.44	2.44
	Interdendrite	24.90	15.43	13.45	30.44	15.78
	Dendrite	23.48	26.00	26.65	23.17	0.96
T2	Design value	23.26	23.26	23.26	23.26	6.98
	Interdendrite	24.81	15.37	13.13	29.89	16.80
	Dendrite	22.74	25.87	26.58	21.15	3.66
T3	Design value	22.22	22.22	22.22	22.22	11.11
	Interdendrite	24.64	15.13	12.87	29.12	18.24
	Dendrite	21.75	24.87	26.32	18.91	8.15
T4	Design value	21.28	21.28	21.28	21.28	14.89
	Interdendrite	24.43	14.89	12.61	28.65	19.42
	Dendrite	20.34	23.89	25.32	17.35	13.10

**Table 3 materials-13-02209-t003:** Electrochemical parameters of different HEAs in 3.5 wt.% NaCl solution.

Alloy	i_corr_ (j/(A/cm^2^))	E_corr_ (V)	∆E (V)
CoCrFeNiTi_0.1_	5.64 × 10^−6^	−1.18	0.87
CoCrFeNiTi_0.3_	4.23 × 10^−6^	−1.16	1.01
CoCrFeNiTi_0.5_	3.81 × 10^−6^	−1.13	1.23
CoCrFeNiTi_0.7_	1.04 × 10^−6^	−1.11	1.34
CoCrFeNiNb [12]	7.23 × 10^−6^	−0.37	0.63
CoCrFeNiW [28]	1.42 × 10^−5^	−0.78	0.62
CoCrFeNiCu [29]	1.77 × 10^−5^	−0.84	1.14

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
