# Peer review of "Effect of Ti Content on the Microstructure and Corrosion Resistance of CoCrFeNiTix High Entropy Alloys Prepared by Laser Cladding"

_materials, 2020, doi:10.3390/ma13102209_

Round 1

Reviewer 1 Report

The manuscript presents results of laser cladding the Q235 steel by high entropy alloys. The influence of different Ti contents in the system CoCrFeNiTix is investigated with regard to corrosion resistance using current density potential measurement under NaCl electrolyte influence. For this topic, a fundamental interest of the readers can be assumed. However, the thematic classification in the introduction part is very broad. There must be a stronger focus on the core topic of coating and corrosion behavior. This requires the inclusion of additional literature. For this purpose, articles from various Special Issues can be used:

https://www.mdpi.com/journal/metals/special_issues/HEA

https://www.mdpi.com/journal/coatings/special_issues/HEA-coatings

https://www.mdpi.com/journal/entropy/special_issues/High-Entropy_Alloys

https://www.mdpi.com/journal/entropy/special_issues/High-Entropy_Materials

In order to ensure reproducibility and to be able to evaluate the coating process, the powders as starting materials must be examined in detail.

In general:

- Unify: High entropy alloys - High-entropy alloys (Keywords)

- Please change superscript literature references according to mdpi standard.

- Generalizing statements should be avoided. 

Abstract

The first sentence in the abstract should be in the introduction part. Is the statement "with synchronous powder feeding" in the abstract relevant? An introductory statement about the investigated molar content of Ti in the abstract would be helpful. A generalization by the statement "Compared with other HEAs, CoCrFeNiTix HEAs have a higher corrosion potential and a lower corrosion current density" should be avoided. A final sentence on the general added value of the presented study would be helpful.

Introduction

"As the number of elements in an alloy is increased, brittle intermetallic phases can be easily formed" is too boldly formulated. More scientific would be a reference to entropy and precipitation kinetics. "Alloys containing these phases show extremely high grain boundary brittleness, resulting in a decline in the mechanical properties." This statement also has no general validity. In the field of nickel-base superalloys, coherent precipitates are specifically used to improve the mechanical properties in the high-temperature range. The related conclusion "High entropy alloys (HEAs) are employed to solve this problem. is not reasonable. The effect of the four core effects has now been partly disproved, therefore the following sentence should also be amended (DOI: 10.1007/s11837-017-2527-z) "Due to the thermodynamic and dynamic four core effects inherent to HEAs, they tend to form a stable solid solution phase and have excellent properties compared to traditional alloys". CoCrFeNi has an FCC structure and is therefore not suitable for cutting materials. However, due to its poor mechanical strength, the CoCrFeNi HEA system cannot fulfil the requirements of high-strength parts, such as cutting tools. What does "ordered phase" mean? fixed stoichiometry? Crystal structures also have an ordering structure. In the next sentence is written "resulting in precipitation strengthening and improvement of the mechanical properties". Initially it was stated that intermetallic phases are worsen the mechanical properties "containing these phases show extremely high grain boundary brittleness". This statement does not fit together and should be changed. "can reduce the cost" requires further explanation. The section starting with "Computational materials science" does not fit the topic presented or the intersections are not well worked out. "Computational materials science" is also not included in the main part. Therefore, this section should be omitted in favor of a more comprehensive presentation of HEA coatings.

In general, a strong focus is put on the mechanical properties and the influence of Ti, Nb and Mo is emphasized. No differentiation is made between the individual alloying elements. However, the broad focus on mechanical properties and computational materials science misses the main topic.

In the experimental part, manufacturer's information on the individual devices must be included. Grams and seconds must be separated by a space. "impurities on the substrate surface were removed with sandpaper" is too unspecific. "were prepared by synchronous feeding" Information on the powder feeding system would be helpful. Furthermore, information on the HEA powder is missing. What does vacuum atomization mean? Were single powders produced and then mixed? If so, how can oxidation of pure iron powder be effectively prevented. How was the reactive Ti powder handled? How much powder was fed into the coating process? How many transitions were applied? Where are the corrosion measurements located? Which test area size was investigated? Which potentiometer was used? Were the coatings mechanically reworked in advance of the investigations?

In the results section, the chosen production route must be checked.

The results section should be supported by a powder characterization. This must include at least morphology, grain size and chemical composition. This is the only way to ensure reproducibility of the results. Based on this, the results of the powder characterization must be compared and discussed with those of the layers. The layer analysis must be more specific. In which area the chemical composition was measured. Did element enrichment occur in the interface area? Did crack formation occur. How is the interface zone in the overlap area structured? With the necessary additional information in the experimental part, the results on corrosion behavior can be conclusively processed.

In the discussion part (line 7) Nb suddenly appears in the layer. How can this be? The axes of the diagrams in Figure 6 must be provided with a unit. Figure 7 describes different positions which are not explained anywhere. A classification is therefore not given.

The discussion is very limited and generalizing. Special powders and a special production route are chosen. However, the results should be generally valid. Microstructure and hardness will differ within the layer, depending on the cooling conditions. Therefore, a much more detailed description would be necessary.

Currently, the results provide only limited added value for the scientific community. Significant improvements are necessary to improve the plausibility.

Author Response

The authors would like to thanks so much for your precious time and invaluable comments. We have carefully addressed all the comments. The corresponding changes and refinements made in the revised paper are summarized in the following document,“Response to Reviewer 1 Comments”. we have made serious changes in the newly revised manuscript, and the "Track Changes" function in Microsoft Word is used to facilitate your review

Reviewer 2 Report

A few comments

  1. Inroduction needs some work. Please refer to other highly cited introductions from this journal
  2. Figure 1 is a copy paste image. Please get a higher quality image. The current figure looks really bad
  3. Give details on the manufacturing of the CoCrFeNiTix HEA. How was it done? Using atomizer ? Temperature ? Time ? etc.
  4. "the elemental segregation is likely to be due to spinodal decomposition during cooling" any reference for your claim ?
  5. How was it established that an FCC phase exists ? Its mentioned a lot but no proper proof is provided. 
  6. How can the authors explain the relation between microstrucutre and hardness of T4 ?

Author Response

The authors would like to thanks so much for your precious time and invaluable comments. We have carefully addressed all the comments. The corresponding changes and refinements made in the revised paper are summarized in the following document,“Response to Reviewer 2 Comments”. we have made serious changes in the newly revised manuscript, and the "Track Changes" function in Microsoft Word is used to facilitate your review.

Reviewer 3 Report

This research was aimed to study the effect of Ti content on the microstructure and corrosion resistance of CoCrFeNiTix high entropy alloys prepared by laser cladding. The subject under analysis is relevant and some interesting results were obtained by the authors.

This paper has potential to be considered for publication. However, the following aspects must be addressed by the authors:

1) The state of the art on the addition of Ti to CoCrFeNi HEAs should be presented in the Introduction. Previous works addressing this subject should be presented.

2) The area of the cladded surface should be indicated.

3) The thickness of the coating layer should be indicated.

4) The authors refer:

“The lattice parameter of the CoCrFeNiTix HEAs increases from 3.491 Å to 3.613 Å due to the lattice distortion caused by the addition of Ti.”

The way these values were achieved should be explained.

5) If the tested range of the XR diffraction angle (2θ) was 20º-100º, the 2θ axis of the XRD spectra cannot have a range of 0º-100º (Fig. 1). The 0º-20º part of the range is not valid.

6) It would be interesting if the authors could present macrograph(s) of the transverse cross-section of the deposited layer.

7) In Fig. 1, a magnified image of the peak zone (around 40º) should be illustrated to sample T4 (like it was presented to the other samples). The smaller peaks are not discernible in the presented figure.

8) In the discussion of Fig. 2, the authors refer:

“The interdendritic regions are enriched with Ti and Ni, whereas the dendritic regions are enriched with Cr and Fe. According to the XRD analysis in Fig. 1, the Ti and Ni-rich interdendritic regions are likely composed of intermetallic precipitates, i.e., the hexagonal Co2Ti phase and the rhombohedral Ni2Ti phase. The Cr and Fe-rich dendritic regions are primarily composed of the FCC solid solution phase. For the T1 alloy, where no secondary phase was observed by XRD, the elemental segregation is likely to be due to spinodal decomposition during cooling.”

The authors specify the case of the sample T1, but do not address the sample T2. The Co2Ti and Ni2Ti phases were not identified in this sample too.

9) Were the SEM micrographs presented in Fig. 2 registered in the cross-section of the processed samples? This should be referred by the authors.

10) The magnification of Fig. 2a is quite different from the magnification of the Figs. 2b to 2d. The magnification of these micrographs should be uniform.

11) In Table 1, The authors should clarify the meaning of the region “design”.

12) The decrease in the Co and Ni contents in the dendritic regions from T1 to T4 should be discussed and related to the detected phases.

13) The authors refer:

“As the Ti content increases, the peak hardness of CoCrFeNiTix HEA coatings increases from 310 Hv to 750 Hv, which is significantly higher than the hardness of the substrate (122 Hv).”

The hardness peak for the sample T4 seems to be higher than 750 HV.

Author Response

The authors would like to thanks so much for your precious time and invaluable comments. We have carefully addressed all the comments. The corresponding changes and refinements made in the revised paper are summarized in the following document,“Response to Reviewer 3 Comments”. we have made serious changes in the newly revised manuscript, and the "Track Changes" function in Microsoft Word is used to facilitate your review

Round 2

Reviewer 1 Report

Thank you for considering the suggestions in the revised manuscript version. There are still some errors in the text that need to be corrected. In addition, some content adjustments are still necessary.
There is a problem with the references in the text. It is no longer possible to assign the bibliographical references. Please insert the literature references without superscript. Partial errors occur. An author named Martin is not included in the bibliography.
Vacuum atomization is a wrong designation. There is inert gas or water atomization which can be performed in vacuum. Vacuum atomization milling is completely misleading. Here, melting metallurgical production is mixed with mechanical alloying. Argon can be a protective gas in high energy ball milling or an atomizing agent. It is still not clear whether pure powders were produced by inert gas atomization and then blended by high energy ball milling. Was a pre alloy atomized? If high-energy ball milling was used, information on the additive (example: steric acid) as well as the balls used and their diameter is required. This part must be revised.
Use the same "x" for geometric specifications. Also insert a space before "mm" (example: 10 mm × 10 mm × 10 mm).
Figure 1 shows a gas atomized powder. This is not a mechanically alloyed powder. Measuring bars are inconsistent and sometimes not visible. 50-60 µm grain size fraction is very narrow. How was this determined?
The measuring bars of all figures should be standardised and changed to a legible size.
Please submit the document without any traceable changes next time. Just mark your additions.
Since there are still significant weaknesses, the manuscript will be subject to major revision.

Author Response

The authors would like to thanks so much for your precious time and invaluable comments. We have carefully addressed all the comments. The corresponding changes and refinements made in the revised paper are summarized in the following document,“Response to Reviewer 1 Comments”. we have made serious changes in the newly revised manuscript, and the "Track Changes" function in Microsoft Word is used to facilitate your review.

Reviewer 3 Report

Considering the changes conducted by the authors, the reviewer thinks that the present paper can be accepted for publication.

Author Response

Thank you very much for your attention and kind comments on our paper.